# FeS_2_/C Nanowires as an Effective Catalyst for Oxygen Evolution Reaction by Electrolytic Water Splitting

**DOI:** 10.3390/ma12203364

**Published:** 2019-10-15

**Authors:** Kefeng Pan, Yingying Zhai, Jiawei Zhang, Kai Yu

**Affiliations:** 1School of Metallurgy, Northeastern University, Shenyang 110819, China; xiaopandy@126.com (K.P.); zhang416940558@163.com (J.Z.); yuk@smm.neu.edu.cn (K.Y.); 2Computing Center, Northeastern University, Shenyang 110819, China

**Keywords:** oxygen evolution reaction, electrocatalyst, FeS_2_/C nanomaterial, electrochemical activation, water splitting

## Abstract

Electrolytic water splitting with evolution of both hydrogen (HER) and oxygen (OER) is an attractive way to produce clean energy hydrogen. It is critical to explore effective, but low-cost electrocatalysts for the evolution of oxygen (OER) owing to its sluggish kinetics for practical applications. Fe-based catalysts have advantages over Ni- and Co-based materials because of low costs, abundance of raw materials, and environmental issues. However, their inefficiency as OER catalysts has caused them to receive little attention. Herein, the FeS_2_/C catalyst with porous nanostructure was synthesized with rational design via the *in situ* electrochemical activation method, which serves as a good catalytic reaction in the OER process. The FeS_2_/C catalyst delivers overpotential values of only 291 mV and 338 mV current densities of 10 mA/cm^2^ and 50 mA/cm^2^, respectively, after electrochemical activation, and exhibits staying power for 15 h.

## 1. Introduction

With the exhaustion of traditional fossil fuels, various economy and ecology issues become serious and need to be resolved [1,2]. Developing green and renewable energy is a promising strategy to solve this problem [3]. The electrochemical splitting of water, converting water into H_2_ and O_2_, has been considered as an environmentally friendly and cost-efficient alternative to traditional energy systems [4]. However, it still has an unavoidable energy loss owing to the high overpotential for the overall reaction of both the hydrogen evolution reaction (HER) and the oxygen evolution reaction (OER) [5]. Furthermore, the second is regarded as the rate-limiting step owing to the sluggishness of its four-electron reaction [6,7]. Therefore, exploring efficient catalysts for the OER process is the key step for improving the overall reaction.

Noble metal-based catalysts exhibit good OER activity, but cost and limited supplies limit the extent of their applications [8]. Different methods have been proposed to create stable, active, and low-cost electrocatalysts. Compounds of transition metals and non-metals of groups have been used for electrocatalysis of OER because they are abundant and have low cost and high activity [6,8,9,10,11,12,13]. For example, the WSe_2_/MoS_2_ heterostructure [14] and Ni- and Mo-based bimetallic metal organic framework [15] have been proven to have good catalytic activity of hydrogen evolution. The combination of CeO_x_ and NiFe–OH can accelerate the electroadsorption energies between the electrocatalyst surface and oxygen intermediates, considerably contributing to enhancement of the OER [16]. However, Fe-based materials are rarely studied when compared with Co- and Ni-based materials, even though they are more environment-friendly and abundant in reserves. Usually, a dual metal-based catalyst with slight Fe addition would exhibit a more enhanced performance toward OER than that only using one host phase [5,17]. The less active sites and intrinsically low electrical conductivity of Fe-based catalysts cause the lower activity of the OER process. Recently, however, many reports demonstrated that Fe is also a good candidate for the active site towards OER [10,18,19]. Operando X-ray absorption spectroscopy was used by Bell and co-workers to study OER over nickel–iron oxyhydroxides [20]. On the basis of the computational results, it was found that Fe^3+^ in Ni_1−x_Fe_x_OOH exhibits short Fe−O bond distances, leading OER intermediates to a nearly optimal binding energy at the Fe sites. The results confirmed that Fe species, not Ni sites, are the active sites for OER [21].

Beside the active site of metal ions, the related compounds are also important for the OER process. Iron-based sulfides have various phases, including FeS [22], FeS_2_ [23], and Fe_3_S_4_ [24], which have large application prospects in the electrochemical and catalytic field. A very abundant sulfur mineral, pyrite (FeS_2_), shows similar electronic properties compared with NiS_2_, and high reactivity in lithium-ion batteries, electrochemical glucose sensors, and photocatalysts [18,25]. However, the activity for the OER process is not good owing to the low conductivity and few active sites. Thus, we need to optimize the catalytic activity of the FeS_2_ catalyst by controlling the structure, size, and crystallinity of the active component. For example, the electrocatalytic activity of FeS_2_/CoS_2_ nanosheets can be significantly improved by producing sulfur vacancies on the interface of these nanosheets [26]; the two-layer structure of porous FeS_2_ coupled with titanium dioxide nanotubes has a good catalytic activity for photochemical water decomposition not only in the ultraviolet and visible regions, but also in the infrared region [27]; the 2D FeS_2_ disc nanostructures have been proved as an efficient and stable hydrogen evolution electrocatalyst, generating hydrogen for up to 125 hours [28]; and FeS_2_ nanoparticles embedded between graphene oxide can significantly improve the catalytic activity of the hydrogen evolution reaction [29].

Herein, we synthesized FeS_2_/C nanowires from FeS_2_–ethylenediamine nanowires. The decomposition of organic groups ensures that the porous structure and carbon conductivity layer encapsulation provide more active sites and higher electron transfer toward the OER process. Especially, the FeS_2_/C nanowires show good performance toward the oxygen evolution reaction after activation. Current densities of 10 mA/cm^2^ and 50 mA/cm^2^ gave overpotentials of only 291 mV and 338 mV, respectively, after the electrochemical activation.

## 2. Materials and Methods

### 2.1. Materials Preparation

FeS_2_/C catalysts were calcined synthesized from a nanowire precursor. FeCl_2_ 4H_2_O (CAS: 13478-10-9, Aladdin, 0.298 g) and polyvinylpyrrolidone (PVP, CAS: 9003-39-8, Sigma Aldrich, Shanghai, China, M_w,avg_ = 40,000, 2.1 g) were dissolved in a deionized water/ethylene glycol (CAS:107-21-1) mixture (15 ml, 1:2 vol ratio) by stirring for 1 h, resulting in solution A. Solution B was prepared by dissolving sulfur powder (CAS:7704-34-9, Aladdin, 0.384 g) in ethylenediamine (CAS: 107-15-3, Beijing Chemical Works, Beijing, China, 10 ml,). Solution B was added to solution A and stirred for 10 h. Then, the mixture was transferred to a Teflon-lined autoclave and kept at 200 °C for 24 h. This mixture was taken to room temperature, and the precipitate (FeS_2_–ethylenediamine) was washed with deionized water and absolute ethanol (CAS: 64-17-5), and then dried overnight at 80 °C under vacuum. The solid was heated in a tubular furnace at 350 °C under N_2_ for 30 min, producing the FeS_2_/C nanowires.

### 2.2. Materials Characterization

Scanning electron microscopy (SEM) was performed on a Hitachi S-4800 field emission SEM (HITACHI, Tokyo, Japan). Thermogravimetric analysis (TGA) curve was characterized on a STA 449 C Jupiter (NETZSCH, Selb, Bavaria, Germany) thermogravimetry analyzer under N_2_ atmosphere. Transmission electron microscopy (TEM) and high-resolution TEM (HRTEM) characterizations were done with a FEI Tecnai G2 F20 instrument (FEI, Hillsboro, OR, USA). Powder X-ray diffraction (XRD) patterns were determined by a Bruker D8 Focus powder X-ray diffractometer (Bruker, Karlsruhe, Baden-Württemberg, Germany) at an operation voltage of 40 kV. Attenuated total reflectance-Fourier transform infrared (ATR-FTIR) spectra were taken at ambient temperature with a FTIR spectrometer (Nicolet Magna, IR 560, Madison, WI, USA). X-ray photoelectron spectroscopy (XPS) was obtained on a Thermo Scientific ESCALAB 250Xi spectrometer (Thermo ScientificWaltham, MA, USA). N_2_ adsorption–desorption measurements were conducted on a Micromeritics ASAP 2010 instrument (Micromeritics Instrument Corporation, Norcross, GA, USA).

### 2.3. Electrochemical Determinations

The OER determinations were done with a BioLogic VMP3 station. For the preparation of the working electrode, 4 mg of the catalyst (FeS_2_/C nanowirdes and commercial IrO_2_ catalyst) was firstly dispersed into a solution prepared with 450 μL of ethanol, 450 μL of DI water, and 100 μL of a (5% by weight) Nafion (CAS: 170006-88-9) solution to make a slurry. Afterward, the slurry was ultrasonicated for 4 h. Then, 10 μL of this homogenized ink was dropped over a glassy carbon electrode (D = 3 mm), which was polished by Al_2_O_3_, and was then left to dry naturally. The electrolyte was 1 M KOH (CAS: 1310-58-3) solution. The counter electrode was a Pt plate and the reference was a Hg/HgO electrode. All potentials in this work are reported versus the reversible hydrogen electrode (RHE) in the working pH, unless otherwise stated. The potentials were converted into the RHE scale according to the below equation:E_RHE_ = E_Hg/HgO_ + E_0 Hg/HgO_ + 0.059 × pH,(1)
where E_RHE_ is the potential on the RHE scale, E_Hg/HgO_ is the potential applied experimentally, and E_0_ is the standard potential of the Hg/HgO redox couple on the normal hydrogen electrode scale (0.098 V). The Tafel slopes were calculated from the overpotential versus log (j) curves for the linear sweep voltammetry (LSV). Both the potential and current accuracy of BioLogic VMP3 station are 0.2% of the current range.

## 3. Results and Discussion

### 3.1. Determination of Properties of the FeS_2_/C

As shown in Figure 1, the synthesis process of FeS_2_/C catalysts mainly consists of two steps. Firstly, FeS_2_–ethylenediamine nanowires are synthesized by a hydrothermal method. In this solvothermal reaction, FeCl_2_·4H_2_O and sulfur powder are used as the Fe and S sources, respectively. Ethylenediamine and ethylene glycol are used as both ligand and solvent, and PVP is a good surfactant to tune the structure and morphology of the composite. Later on, FeS_2_/C nanowires are readily prepared by pyrolysis of the precursor. The decomposition of the organic molecules generates pores in the carbon shells of the nanowires.

The morphology of the as-prepared FeS_2_–ethylenediamine precursor are determined by electron microscopy. The SEM (Figure 2a) image reveals that the FeS_2_–amine nanowires (d ≈ 100 nm) and lengths up to tens of micrometers are successfully synthesized. The precursor is featured with a highly uniform nanowire morphology and a fairly smooth surface. The mixture solvent of ethylenediamine and ethylene glycol, which have linear configuration, are served as template molecules and induced the growth of the nanostructure during the solvothermal process [30,31]. Especially, ethylenediamine binds strongly to Fe ions [32,33]. PVP is frequently used for chemical reduction or for controlling the three-dimensional structure of the products. The lone electron pairs of oxygen on the PVP can give away, and coordinated with Fe ions in the reaction [34]. A 5 nm amorphous and uniform outer layer is shown by TEM (Figure 2b). This outer layer derives from the surfactant PVP. In order to verify the effect of PVP on the structure of the composite, the precursor without the addition of PVP is synthesized. As shown in Appendix A, the precursor was composed by nanowires and bulk particles. The width of nanowires in this sample was much larger than that of the FeS_2_–ethylenediamine precursor with the addition of PVP. The TEM image in Appendix A shows that the edge of FeS_2_–ethylenediamine is smooth and without any layer encapsulation. The content of the outer layer is also verified by the characteristic absorption peaks in the FTIR spectra of the precursor FeS_2_– ethylenediamine nanowires (Appendix A). The strong ν_as_(CO) mode at 1648 cm^−1^; ν(CC) mode at 1114 cm^−1^; and ν_as_(CH) and ν_s_(CH) stretch modes at 2932 and 2866 cm^−1^, respectively, indicate the existence of PVP [35].

Figure 3 shows the TGA curve of the FeS_2_–ethylenediamine precursor under an N_2_ atmosphere between 20 and 800 °C. The small loss at the start (below 100 °C) is because of the loss of water. It starts to dramatically lose weight at around 120 °C until reaching the temperature of 300 °C, with a loss of 42.3%, attributable to the evaporation or decomposition of organic content. Only 12.6% weight loss is observed from 350 °C to 800 °C due to the further transition of pyrite FeS_2_. Thus, 350 °C is chosen as the calcination temperature to obtain FeS_2_.

In order to confirm the phase of FeS_2_ after calcination treatment in the N_2_ atmosphere, the composition of the product is determined by XRD. As shown in Figure 4a, the XRD pattern possessed plenty of noises and seems like amorphous; this XRD signature may be typical of mostly amorphous carbon existing at the boundary of the FeS_2_/C nanowires (Figure 4d), and is 3 nm thick with a certain degree of sp^2^-hybridization, resulting in regions of “graphitic” carbon. Except for the broad peak at around 25°, which is derived from the amorphous carbon, all the peaks can be assigned to pyrite FeS_2_ (JCPDS No. 42-1340) [23,32]. No other detectable peaks from impurities (marcasite, greigite FeS_2_, sulfur, or other) are observed in the pattern, indicating the high purity of this as-prepared sample.

The morphology and detailed structures of the as-prepared FeS_2_/C product were further investigated by SEM and TEM. After calcination, the nanowires’ morphology is essentially preserved during the calcination process. However, the highly magnified SEM image (Figure 4b) reveals that the rough surface and porous structure are observed on the FeS_2_/C nanowires. At the same time, it can also be observed from Figure 4b that the obtained sample seems to not be very regular, owing to the agglomeration of fibers. For this reason, the catalyst was dispersed in the mixed solution of ethanol and water for 4 h under ultrasonic to prepare the catalyst electrode. The ultrasonic process effectively ensured the reproducibility of the catalytic performance. TEM and the high-resolution TEM (HRTEM) were carried out to further confirm the inner microstructure and the crystallographic structure of the FeS_2_/C nanowires. As indicated in Figure 4c, the FeS_2_/C nanowires are of a porous structure with a diameter of about 220 nm. The porous structure is derived from the decomposition of organic groups in the organic–inorganic hybrid FeS_2_–ethylenediamine precursor. The HRTEM image in Figure 4d shows a 3 nm thick, uniform, amorphous layer at the boundary of the FeS_2_/C nanowires. Furthermore, a layer spacing of 0.27 nm corresponds to the interplane spacing of the pyrite FeS_2_ (200) plane according to XRD results, indicating the successful synthesis of FeS_2_.

The XPS measurement is carried out to examine the elemental compositions and atomic bonding states of FeS_2_. The high-resolution XPS of Fe and S of FeS_2_/C nanowires were deconvoluted, and the results are shown in Figure 5. Figure 5a shows the Fe 2p_3/2_ and Fe 2p_1/2_ peaks of Fe at 708.5 and 723.5 eV, respectively. Figure 5b shows the XPS of S 2p of FeS_2_/C nanowires. The two peaks centered at 159.3 and 160.5 eV should be assigned to S 2p_3/2_ and S 2p_1/2_, respectively [36]. The S 2p presented in Figure 5b shows a peak around 165.2 eV, characteristic of the sulfate species, which may be because of the presence of the FeSO_4_ phase. Additionally, it has been demonstrated that FeSO_4_ species are produced on the surface of FeS_2_ when it is exposed to water [37].

The porous structure of FeS_2_/C nanowires is characterized by the N_2_ adsorption–desorption measurement. As shown in Appendix A, the nitrogen adsorption–desorption isotherms correspond to a type IV curve with a distinct hysteresis loop. The pore-size distribution curve (Appendix A), according to the BJH (Barrett-Joyner-Halenda) method, presents the distinct mesoporous microstructure in FeS_2/_C nanowires with a broad pore size.

### 3.2. Electrochemical Oxygen Evolution Reaction

The OER activity of FeS_2_/C catalysts was investigated in a 1.0 M KOH aqueous solution. The working electrode was made of glassy carbon covered with FeS_2_/C nanowires, the counter electrode was a Pt plate, and the reference was the Hg/HgO electrode. LSV scans were performed on the commercial IrO_2_ as the benchmark and FeS_2_/C catalysts at different scan cycles (Figure 6a). The early onset and the large anodic current of all samples indicate their high OER catalytic activity. Figure 6a shows that the working electrode has a good catalytic activity for OER at the first cycle, which is better than the pure iron sulfides without doping other metal ions reported before [19]. The porous structure provides more sites for the OER reaction, and the encapsulation of carbon enhances the conductivity of the catalyst toward a fast electron transfer. The catalyst performance of the FeS_2_/C nanowires is still slightly worse than that of the commercial IrO_2_ catalyst. However, the catalytical activity of the FeS_2_/C nanowires enhanced with the cycling, owing to the activation of FeS_2_. The FeS_2_/C sample has a reduced potential of 1.58 V, an overpotential of 350 mV, and a current density of 10 mA/cm^2^ at the 20th cycle, which continue reducing until the 80th cycle. Furthermore, an impressively low onset potential of 257 mV and overpotential values of only 291 mV and 338 mV were obtained for the current densities of 10 mA/cm^2^ and 50 mA/cm^2^, respectively, at the 100th cycle, which are much lower than those for the first cycle, indicating the succesful activation process of FeS_2_ during cycling. Furtermore, the CV curves of FeS_2_/C nanowires were also tested in 1 M KOH at various scan rates. As shown in Appendix A, when the catalyst scans at a high scan rate, such as 10, 50, and 100 mV/s, all the CV curves exhibit a pair of redox peaks with a redox potential of about 1.42 V, which is assigned to the redox reaction between Fe^2+^ and Fe^3+^.

Meanwhile, the reaction kinetics and the activity of the FeS_2_/C catalyst are also confirmed by the Tafel slopes. As shown in Figure 6b, these curves show linear portions that are adjusted to the equation η = b log j + a (Tafel Equation), wherein η is the overpotential (V, refers to the theoretical OER potenital of 1.23 V), j is the current density (mA/cm^2^), and b is the Tafel slope. The FeS_2_/C electrode shows a high Tafel slope of about 84.9 mV/dec at the first cycle, which reduced to 65.6 mV/dec at the 100th cycle.

The detailed mechanism of the FeS_2_/C catalyst toward OER is not well understood, however, we believe that it is similar to that of metal oxide electrodes. In the alkaline environment, the OER proceeds as described in the following Equations:(2)$$4OH^{-}\to OH^{*}+3OH^{-}+e^{-},$$
(3)$$OH^{*}+3OH^{-}\to O^{*}+2OH^{-}+H_{2}O+e^{-},$$
(4)$$O^{*}+2OH^{-}+H_{2}O\to OOH^{*}+OH^{-}+H_{2}O+e^{-},$$
(5)$$OOH^{*}+OH^{-}+H_{2}O\to O_{2}+2H_{2}O+e^{-}.$$

In this mechanism, the intermediates *OH**, *O**, and *OOH** are formed in thermodynamically costly processes. The step that determines the activity of the catalyst is the one with the highest energy barrier and is called the rate-limiting step. From the results shown above, the second step could be considered rate-determining for the OER in alkaline medium. In this case, Fe (II) in FeS_2_ suffers a transition to Fe (III) that could be part of the catalytic site of the catalyst. It has been published that mackinawite (FeS) grown on iron foam was highly active for oxygen evolution. This material, when subjected to the OER, forms an FeO_x_ film by oxidation-desulfuration. Thus, we assumed that FeS_2_/C was partially oxidized, liberating SO_4_^2−^ [18,38]. The OER charge-transfer kinetics of the catalyst was further studied by electrochemical impedance spectroscopy in 1 M KOH. As shown in Appendix A, both the Nyquist plot before and after the OER test show a semicircle, the value of which serves as the function of electron-transfer resistance on the electron surface. The larger semicircle of FeS_2_/C before the OER test reflects a higher charge transfer resistance on the interface of the electrocatalyst. Remarkably, FeS_2_/C after the OER test shows a much smaller semicircle, reflecting a significantly enhanced conductivity with a lower charge transfer resistance and more rapid catalytic kinetics, and verifying the enhanced OER performance.

To evaluate the electrocatalytic durability of the FeS_2_/C catalyst, chronopotentiometric measurements towards the OER were performed at 10 mA/cm^2^. As shown in Figure 7, the potential of the FeS_2_/C catalyst decreases at the beginning, owing to the activation of FeS_2_. Furthemore, the FeS_2_/C catalyst stayed stable for 15 h without an appreciable change in overpotential. Thus, these results confirm that FeS2/C is a promising catalyst in alkaline 10 mA/cm^2^ solutions. Surprisingly, the as-prepared FeS_2_/C nanowires presented a superior or comparable activity to many recently reported OER catalysts (Appendix A). For example, Ni/Mo_x_C (overpotential_@10mA/cm2_ = 328 mV, Tafel slope = 74 mV/dec) [39], Fe_3_C@NCNT/NPC (overpotential_@10mA/cm2_ = 339 mV, Tafel slope = 62 mV/dec) [40], γ-MoC/Ni@NC (overpotential_@10mA/cm2_ = 310 mV, Tafel slope = 62.7 mV/dec) [41], Fe_3_C@NG-800 (overpotential_@10mA/cm2_ = 361 mV, Tafel slope = 62 mV/dec) [42], FeNiS_2_ NSs (overpotential_@10mA/cm2_ = 310 mV, Tafel slope = 46 mV/dec) [43], and CP/CTs/Co-S (overpotential_@10mA/cm2_ = 306 mV, Tafel slope = 72 mV/dec) [44].

## 4. Conclusions

In this work, large-scale FeS_2_/C nanowires were synthesized from FeS_2_–ethylendiamine precursors with rational design via the in situ electrochemical activation method and served as an electrocatalyst toward OER in 1 M KOH. The sluggish kinetics of the oxygen evolution reaction is accelerated by in situ electrochemical activation of the FeS_2_/C nanowires on the electrode by scanning few cycles. The as-prepared FeS_2_/C catalyst demonstrated superior catalytic activity with an ultralow Tafel slope of 65 mV/dec and a low overpotential of 291 mV at 10 mA/cm^2^. The porous nanowire structure provides a good contact between the active Fe (III) sites and the electrolite and explains the good performance. The as-prepared FeS_2_/C nanowires presented a superior or comparable activity to many recently reported OER catalysts. We believe that this work is a foundation for further work about new effective, inexpensive metal-sulfide electrocatalysts, useful for many electrochemical applications.

## 5. Patents

There are no patents resulting from the work reported in this manuscript.

## Figures and Tables

**Figure 1 materials-12-03364-f001:**
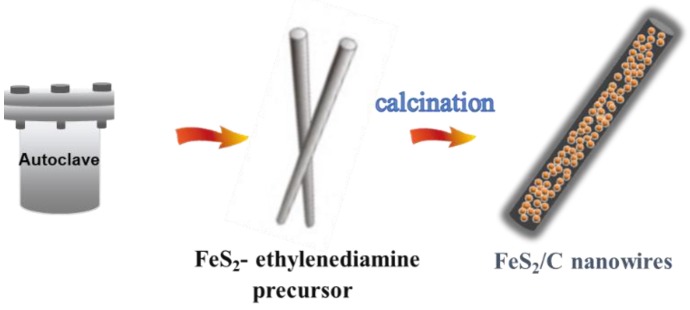
Synthesis of FeS_2_/C nanowires.

**Figure 2 materials-12-03364-f002:**
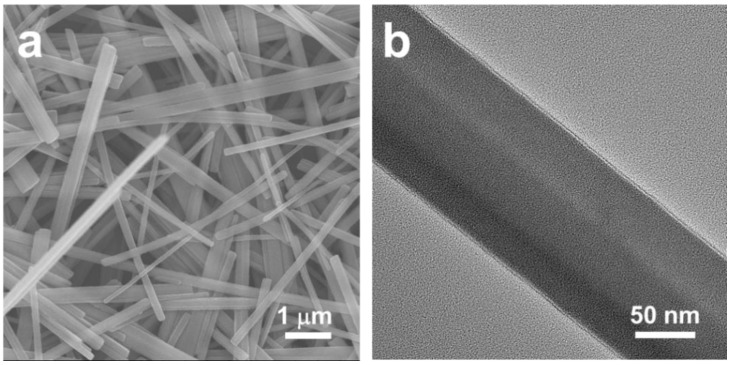
(**a**) Scanning electron microscopy (SEM) and (**b**) transmission electron microscopy (TEM) images of the FeS_2_–ethylenediamine precursor.

**Figure 3 materials-12-03364-f003:**
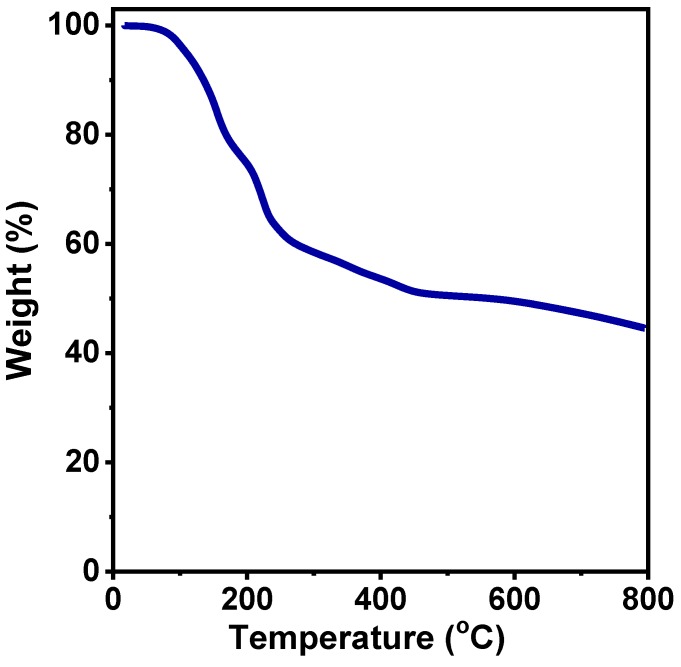
Thermogravimetric analysis (TGA) of the FeS_2_–ethylenediamine precursor.

**Figure 4 materials-12-03364-f004:**
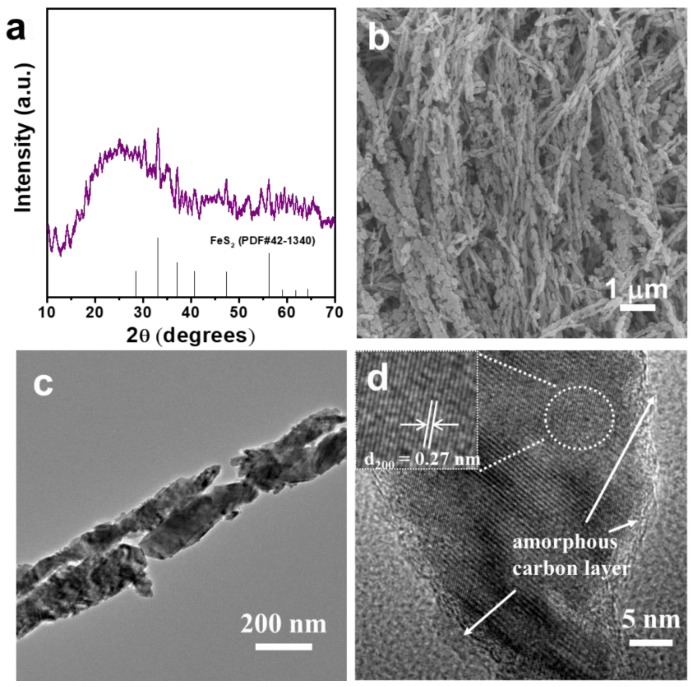
(**a**) X-ray diffraction (XRD) pattern, (**b**) SEM image, (**c**) TEM image, and (**d**) high-resolution TEM (HRTEM) image of as-prepared FeS_2_/C nanowires.

**Figure 5 materials-12-03364-f005:**
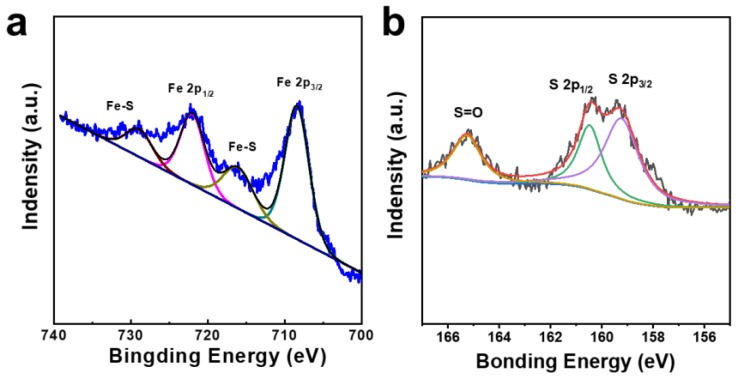
High-resolution X-ray photoelectron spectroscopy (XPS) of (**a**) Fe and (**b**) S in as-prepared FeS_2_/C nanowires.

**Figure 6 materials-12-03364-f006:**
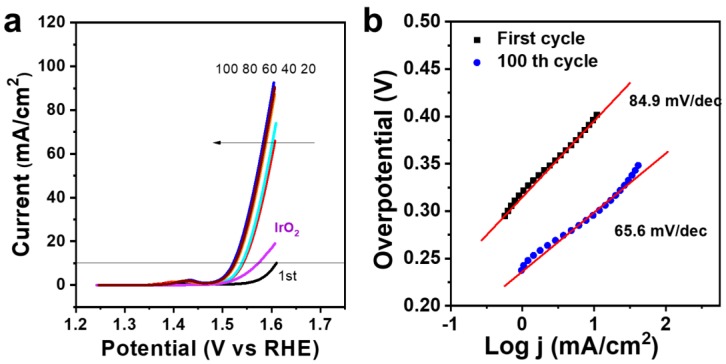
(**a**) Linear sweep voltammetry (LSV) curves and (**b**) Tafel plots of the FeS_2_/C catalyst at different cycles and the comparison with the IrO_2_ catalyst. Scan rate-1 mV/s. RHE, reversible hydrogen electrode.

**Figure 7 materials-12-03364-f007:**
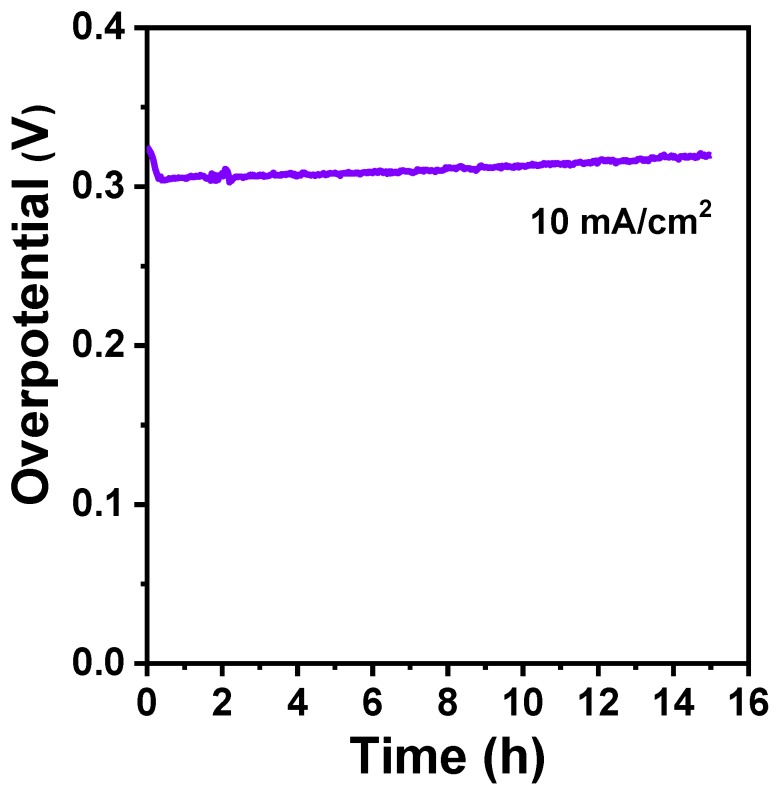
Chronopotentiometric measurement for the long-term stability test of the FeS_2_/C catalyst at the current densities of 10 mA/cm^2^.

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
