# Peer review of "FeS_2_/C Nanowires as an Effective Catalyst for Oxygen Evolution Reaction by Electrolytic Water Splitting"

_materials, 2019, doi:10.3390/ma12203364_

Round 1
Reviewer 1 Report
The manuscript deals with the catalyst based on iron sulfide for oxygen evolution by electrolytic water splitting. The results are clearly stated in the manuscript, but there is a minor remark:
Comments
1). In title of manuscript the name of the test process «electrolytic water splitting» should be added. More correct: «…for Oxygen Evolution by Electrolytic Water Splitting».
2). In Keywords the «water splitting» should be added.
3). The Section 2 «Materials and Methods» should be placed before the Conclusions - rules of the publisher.
4). CAS or Product Number should be given for each of the compounds used in the experiments (in Section «Materials and Methods»).
5). Page 3, Line 98: the symbol of «S» should be replaced by the words of «sulfur powder».
6). What is the difference between «FeS2-ethylenediamine precursor» and «FeS2-amine precursor»? Uniformity in designation of samples should be maintained.
7). What is the error of experimental measurements?
8). To evaluate the results of the study, I recommend comparing them with world achievements. The results will be better understood!!!
9). The author must carefully check the manuscript so that there are no spelling or grammatical mistakes.
Best regards, Reviewer.
Author Response
Dear Reviewer:
Thank you for your comments concerning our manuscript originally entitled “FeS2/C Nanowires as An Effective Catalyst for Oxygen Evolution Reaction” (ID: materials-602695). Those comments are all valuable and very helpful for revising and improving our paper, as well as the important guiding significance to our researches. We have studied comment carefully and have made correction which we hope meet with approval. Revised portion are marked using the "Track Changes" function in Microsoft Word in the paper. The main corrections in the paper and the responds to the reviewer’s comments are as flowing:
1.Comment: In title of manuscript the name of the test process «electrolytic water splitting» should be added. More correct: «…for Oxygen Evolution by Electrolytic Water Splitting».
Response: Thanks for your comment, the title of the manuscript has been revised. The new title of the manuscript has been revised to “FeS2/C Nanowires as An Effective Catalyst for Oxygen Evolution Reaction by Electrolytic Water Splitting”
2.Comment: In Keywords the «water splitting» should be added.
Response: Thanks for your comment, the commented keyword has been added. The key words of the manuscript have been revised to “oxygen evolution reaction; electrocatalyst; FeS2/C nanomaterial; electrochemical activation; water splitting”
3.Comment: The Section 2 «Materials and Methods» should be placed before the Conclusions - rules of the publisher.
Response: Thanks for your comment, We prepared the manuscript according to the requirements of the journal template document. The order of each section is as follows: Title, Abstract, Keywords, 1. Introduction, 2. Materials and Methods, 3. Results and Discussion, 4. Conclusions, 5. Patents, Author Contributions, Funding, Conflicts of Interest and References.
4.Comment: CAS or Product Number should be given for each of the compounds used in the experiments (in Section «Materials and Methods»).
Response: Thanks for your comment, the CAS number of the compounds used in the experiments have been given in the revised manuscript.
5.Comment: Page 3, Line 98: the symbol of «S» should be replaced by the words of «sulfur powder».
Response: Thanks for your comment, the symbol of “S” has been replaced by the words of “sulfur powder”.
6.Comment: What is the difference between «FeS2-ethylenediamine precursor» and «FeS2-amine precursor»? Uniformity in designation of samples should be maintained.
Response: Thanks for your comment, the sample name has been unified into “FeS2-ethylenediamine precursor”.
7.Comment: What is the error of experimental measurements?
Response: Thanks for your comment, the error of experimental measurements has been added.
“Both the potential and current accuracy of BioLogic VMP3 station are 0.2 % of current range.”
8.Comment: To evaluate the results of the study, I recommend comparing them with world achievements. The results will be better understood!!!
Response:Thanks for your suggestion. The comparison has been added in page 8 as shown below:
Surprisingly, the as-prepared FeS2/C nanowires presented a superior or comparable activity to many recently reported OER catalyst. Such as Ni/MoxC (Overpotential@10mA/cm2=328 mV, Tafel slope=74 mV/dec )[37], Fe3C@NCNT/NPC (Overpotential@10mA/cm2=339 mV, Tafel slope=62 mV/dec)[38], g-MoC/Ni@NC (Overpotential@10mA/cm2=310 mV, Tafel slope=62.7 mV/dec)[39], Fe3C@NG-800 (Overpotential@10mA/cm2=361 mV, Tafel slope=62 mV/dec)[40], FeNiS2 NSs (Overpotential@10mA/cm2=310 mV, Tafel slope=46 mV/dec)[41], CP/CTs/Co-S (Overpotential@10mA/cm2=306 mV, Tafel slope=72 mV/dec)[42].
9.Comment: The author must carefully check the manuscript so that there are no spelling or grammatical mistakes.
Response: Thanks for your comment, we have checked the manuscript carefully. The spelling and grammatical mistakes have been revised.
We tried our best to improve the manuscript and made some changes in the manuscript. These changes will not influence the content and framework of the paper. And here we listed the changes. We appreciate for Editors/Reviewers’ warm work earnestly, and hope that the correction will meet with approval.
Once again, thank you very much for your comments and suggestions.
Best regards,Authors.
Reviewer 2 Report
Dear Editor
I accurately reviewed the article material 602695
Title: FeS2/C Nanowires as An Effective Catalyst for Oxygen Evolution Reaction
submitted to Materials.
The authors prepared and studied FeS2/C catalyst with porous nanostructured fibers, which serve catalytic reaction in OER process.
The topic is suitable for the journal, but the manuscript is poor in various aspects.
General concerns
The fibers obtained are not one-dimensional, even if they are very long.
The obtained fibers are not monodisperse or regular, and this should be studied because it could affect the reproducibility of the processes.
After the first and after the second preparation step the product obtained should be well characterized by XPS. Moreover, the porosity should be characterized with PET. Cyclic voltammetry curves in a capacitive current region at various scan rates should be performed.
Finally, it could be useful for the readers a comparison of OER catalytic activity of this material with that of other recently reported metal-based catalysts.
In my opinion the article is not suitable for publication in Materials, but I can suggest Inorganics.
best regards
Author Response
Dear Reviewer:
Thank you for your comments concerning our manuscript originally entitled “FeS2/C Nanowires as An Effective Catalyst for Oxygen Evolution Reaction” (ID: materials-602695). Those comments are all valuable and very helpful for revising and improving our paper, as well as the important guiding significance to our researches. We have studied comment carefully and have made correction which we hope meet with approval. Revised portion are marked using the "Track Changes" function in Microsoft Word in the paper. The main corrections in the paper and the responds to the reviewer’s comments are as flowing:
1.Comments: The fibers obtained are not one-dimensional, even if they are very long.
Response: Thanks for your comment. The attribute of the obtained has been deleted.
2.Comments: The obtained fibers are not monodisperse or regular, and this should be studied because it could affect the reproducibility of the processes.
Response: Thanks for your suggestion. We have added an explanation of this problem to the revised manuscript. “At the same time, it can also be observed from Figure 4b, the obtained sample seems not very regular due to the agglomeration of fibers. For this reason, the catalyst was dispersed in the mixed solution of ethanol and water for 4 h under ultrasonic to prepare the catalyst electrode. The ultrasonic process effectively ensured the reproducibility of the catalytic performance.”
3.Comments: After the first and after the second preparation step the product obtained should be well characterized by XPS.
Response: Thanks for your comment. The High-resolution XPS of Fe and S in as-prepared FeS2/C nanowires and corresponding discussion were added in the manuscript.
XPS measurement is carried out to examine the elemental compositions and atomic bonding states of FeS2. The high-resolution XPS of Fe and S of FeS2/C nanowires have been deconvoluted, and the results are shown in Figure 5. Figure 5a shows that the Fe 2p3/2 and Fe 2p1/2 peaks of Fe at 708.5 and 723.5 eV, respectively. Figure 5b shows that the XPS of S 2p of FeS2/C nanowires. The two peaks with centered at 159.3 and 160.5 eV should be assigned to S 2p3/2 and S 2p1/2, respectively. The S 2p presented in Fig. 5b shows peak around 165.2 eV, characteristic of sulfate species, which may be due to the presence of FeSO4 phase. Additionally, it has been demonstrated that FeSO4 species are produced on the surface of FeS2 when it is exposed to water.
4.Comments: Moreover, the porosity should be characterized with PET.
Response: Thanks for your suggestion. The porous structure FeS2@C nanowires is characterized by N2 adsorption-desorption measurement which are added in the manuscript and supporting information.
The relevant description and analysis were added to the revised manuscript.
The porous structure FeS2@C nanowires is characterized by N2 adsorption-desorption measurement. As shown in Figure S3a, the nitrogen adsorption-desorption isotherms correspond to a type IV curve with a distinct hysteresis loop. The pore-size distribution curve (Figure S3b) according to the BJH method, present the distinct mesoporous microstructure in FeS2@C nanowires with a broad pore size.
5.Comments: Cyclic voltammetry curves in a capacitive current region at various scan rates should be performed.
Response: Thanks for your suggestion. The CV curves in the capacitive current region at various scan rates are added in the manuscript.
The relevant description and analysis were added to the revised manuscript.
Furtermore, the CV curves of FeS2/C nanowires are also tested in 1 M KOH at various scan rates. As shown in Figure S4, when the catalyst scanned at high scan rate, such as 10, 50 and 100 mV/s, all the CV curves exhibit a pair of redox peaks with a redox potential about 1.42 V, which assigned to the redox reaction between Fe2+ to Fe3+.
6.Comments: Finally, it could be useful for the readers a comparison of OER catalytic activity of this material with that of other recently reported metal-based catalysts.
Response: Thanks for your suggestion. The comparison has been added in page 8 as shown below:
Surprisingly, the as-prepared FeS2/C nanowires presented a superior or comparable activity to many recently reported OER catalyst. Such as Ni/MoxC (Overpotential@10mA/cm2=328 mV, Tafel slope=74 mV/dec )[37], Fe3C@NCNT/NPC (Overpotential@10mA/cm2=339 mV, Tafel slope=62 mV/dec)[38], g-MoC/Ni@NC (Overpotential@10mA/cm2=310 mV, Tafel slope=62.7 mV/dec)[39], Fe3C@NG-800 (Overpotential@10mA/cm2=361 mV, Tafel slope=62 mV/dec)[40], FeNiS2 NSs (Overpotential@10mA/cm2=310 mV, Tafel slope=46 mV/dec)[41], CP/CTs/Co-S (Overpotential@10mA/cm2=306 mV, Tafel slope=72 mV/dec)[42].
We tried our best to improve the manuscript and made some changes in the manuscript. These changes will not influence the content and framework of the paper. And here we listed the changes. We appreciate for Editors/Reviewers’ warm work earnestly, and hope that the correction will meet with approval.
Once again, thank you very much for your comments and suggestions.
Best regards, Authors.

Reviewer 3 Report
In the introduction section, the author should provide proper explorations about the HER and OER activities of transition metal chalcogenides electrocatalysts with appropriate recent bibliographies (10.1016/j.apsusc.2019.02.236; 10.1021/acssuschemeng.9b03496; 10.1016/j.apsusc.2019.02.042). Also, the authors have to address how the present electro catalyst system is loftier than other FeS2 electro catalyst in the introduction section (10.1002/smll.201801070; 10.1021/acssuschemeng.6b01533; 10.1021/acscatal.5b01637; 10.1002/aenm.201700482). Figure 4, the XRD pattern possessed plenty of noises and seems like amorphous but the author mentioned “the crystal structure of electrocatalyst” in page number 4, please provide a proper explanation for that. In figure S2, the author should enlarge the electrocatalyst FT-IR spectra, it is difficult to identify the major vibrational peaks. In order to identify the various elements and their corresponding oxidation states (Fe and S), the author should provide XPS spectra in the revised version. As far as concerning the electrocatalyst performance towards OER activities, the author should include the impedance plot for the prepared electrocatalyst in the updated version. The paper claims "high electrochemical performance" for the proposed FeS2 electrocatalyst but there is no comparison table of the performance to the previously reported transition metal sulfides. The conclusion section is too short and floating, it should be modified.
Author Response
Dear Reviewer:
Thank you for your comments concerning our manuscript originally entitled “FeS2/C Nanowires as An Effective Catalyst for Oxygen Evolution Reaction” (ID: materials-602695). Those comments are all valuable and very helpful for revising and improving our paper, as well as the important guiding significance to our researches. We have studied comment carefully and have made correction which we hope meet with approval. Revised portion are marked using the "Track Changes" function in Microsoft Word in the paper. The main corrections in the paper and the responds to the reviewer’s comments are as flowing:
1.Comments: In the introduction section, the author should provide proper explorations about the HER and OER activities of transition metal chalcogenides electrocatalysts with appropriate recent bibliographies (10.1016/j.apsusc.2019.02.236; 10.1021/acssuschemeng.9b03496; 10.1016/j.apsusc.2019.02.042). Also, the authors have to address how the present electro catalyst system is loftier than other FeS2 electro catalyst in the introduction section (10.1002/smll.201801070; 10.1021/acssuschemeng.6b01533; 10.1021/acscatal.5b01637; 10.1002/aenm.201700482).
Response: Thanks for your suggestion, the related explorations have been added in the introduction section.
In the second paragraph of the revised introduction: “For example, WSe2/MoS2 heterostructure[14] and Ni, Mo based bimetallic metal organic framework[15] have been proved to have good catalytic activity of hydrogen evolution. The combination of CeOx and NiFe-OH can accelerate the electroadsorption energies between the electrocatalyst surface and oxygen intermediates, considerably contributing to the OER enhancement[16].”
In the third paragraph of the revised introduction: “For example, The electrocatalytic activity of FeS2/CoS2 nanosheets can be significantly improved by producing sulfur vacancies on the interface of these nanosheets[26], the two-layer structure of porous FeS2 coupled with titanium dioxide nanotubes has a good catalytic activity for photochemical water decomposition not only in the ultraviolet and visible regions, but also in the infrared region[27], the 2D FeS2 disc nanostructures has been proved as an efficient and stable hydrogen evolution electrocatalyst with the generating hydrogen up to 125 hours[28], FeS2 nanoparticles embedded between graphene oxide can significantly improve the catalytic activity of hydrogen evolution reaction[29].”
2.Comments: Figure 4, the XRD pattern possessed plenty of noises and seems like amorphous but the author mentioned “the crystal structure of electrocatalyst” in page number 4, please provide a proper explanation for that.
Response: Thank you for your suggestion. Yes, it is difficult to identify the crystal structure of electrocatalyst according to the XRD pattern, we already revised this part.
As shown in Figure 4a, the XRD pattern possessed plenty of noises and seems like amorphous, this XRD signature may be typical of mostly amorphous carbon existing at the boundary of the FeS2/C nanowires with 3nm thick (Figure 4d), with a certain degree of sp2-hybridization, resulting in regions of "graphitic" carbon.
3.Comments: In figure S2, the author should enlarge the electrocatalyst FT-IR spectra, it is difficult to identify the major vibrational peaks.
Response: Thank you for your suggestion. The enlarged FT-IR spectra already added in Fig S2.
4.Comments: In order to identify the various elements and their corresponding oxidation states (Fe and S), the author should provide XPS spectra in the revised version.
Response: Thanks for your comment. The High-resolution XPS of Fe and S in as-prepared FeS2/C nanowires and corresponding discussion were added in the manuscript.
XPS measurement is carried out to examine the elemental compositions and atomic bonding states of FeS2. The high-resolution XPS of Fe and S of FeS2/C nanowires have been deconvoluted, and the results are shown in Figure 5. Figure 5a shows that the Fe 2p3/2 and Fe 2p1/2 peaks of Fe at 708.5 and 723.5 eV, respectively. Figure 5b shows that the XPS of S 2p of FeS2/C nanowires. The two peaks with centered at 159.3 and 160.5 eV should be assigned to S 2p3/2 and S 2p1/2, respectively. The S 2p presented in Fig. 5b shows peak around 165.2 eV, characteristic of sulfate species, which may be due to the presence of FeSO4 phase. Additionally, it has been demonstrated that FeSO4 species are produced on the surface of FeS2 when it is exposed to water.
5.Comments: As far as concerning the electrocatalyst performance towards OER activities, the author should include the impedance plot for the prepared electrocatalyst in the updated version.
Response: Thanks for your comment. The Nyquist plots of the FeS2/C nanowires before and after OER test has been added in the manuscript.
The relevant description and analysis were added to the revised manuscript.
The OER charge-transfer kinetics of the catalyst was further studied by electrochemical impedance spectroscopy in 1 M KOH. As shown in Fig. S4, both the Nyquist plot before and after OER test show a semicircle which vaule serves as the function of electron-transfer resistance on the electron surface. The larger semicircle of FeS2/C before OER test reflects a higher charge transfer resistance on the interface of electrocatalyst. Remarkably, FeS2/C after OER test shows a much smaller semicircle, reflecting a significantly enhanced conductivity with a lower charge transfer resistance and more rapid catalytic kinetics, and verifying the enhanced OER performance.
6.Comments: The paper claims "high electrochemical performance" for the proposed FeS2 electrocatalyst but there is no comparison table of the performance to the previously reported transition metal sulfides.
Response: Thanks for your suggestion. The comparison has been added in page 8 as shown below:
Surprisingly, the as-prepared FeS2/C nanowires presented a superior or comparable activity to many recently reported OER catalyst. Such as Ni/MoxC (Overpotential@10mA/cm2=328 mV, Tafel slope=74 mV/dec )[37], Fe3C@NCNT/NPC (Overpotential@10mA/cm2=339 mV, Tafel slope=62 mV/dec)[38], g-MoC/Ni@NC (Overpotential@10mA/cm2=310 mV, Tafel slope=62.7 mV/dec)[39], Fe3C@NG-800 (Overpotential@10mA/cm2=361 mV, Tafel slope=62 mV/dec)[40], FeNiS2 NSs (Overpotential@10mA/cm2=310 mV, Tafel slope=46 mV/dec)[41], CP/CTs/Co-S (Overpotential@10mA/cm2=306 mV, Tafel slope=72 mV/dec)[42].
7.Comments: The conclusion section is too short and floating, it should be modified.
Response: Thank you for your comment. It already modified as below:
In this work, large-scale FeS2/C nanowires were synthesized from FeS2-ethylendiamine precursors with rational design via in situ electrochemical activation method and served as electrocatalyst toward OER in 1 M KOH. The sluggish kinetics of the oxygen evolution reaction is accelerated by in situ electrochemical activation of FeS2/C nanowires on the electrode by scanning few cycles. The as-prepared FeS2/C catalyst demonstrated superior catalytic activity with an ultralow Tafel slope of 65 mV/dec and a low overpotential of 291 mV at 10 mA/cm2. The porous nanowire structure provides a good contact between the active Fe(III) sites and the electrolite and explains the good performance. The as-prepared FeS2/C nanowires presented a superior or comparable activity to many recently reported OER catalyst. We Believe that this work is a foundation for further work about new effective, inexpensive metal-sulfide electrocatalysts, useful for many electrochemical applications.
We tried our best to improve the manuscript and made some changes in the manuscript. These changes will not influence the content and framework of the paper. And here we listed the changes. We appreciate for Editors/Reviewers’ warm work earnestly, and hope that the correction will meet with approval.
Once again, thank you very much for your comments and suggestions.
Best regards, Authors.

Round 2
Reviewer 2 Report
Dear Editor
I accurately reviewed the article material 602695-v2
Title: FeS2/C Nanowires as An Effective Catalyst for Oxygen Evolution Reaction by Electrolytic Water Splitting
submitted to Materials.
The authors prepared and studied FeS2/C catalyst with porous nanostructured fibers, which serve catalytic reaction in OER process.
In this revised version of the manuscript, the authors solved a lot of problems and greatly improved the paper.
Only small aspects to be corrected.
Paragraph "3.2. Electrochemical oxygen evolution reaction" must begin with some text that introduces figure 6.
The comparison with other systems, whose specifications are reported in the text (lines 250-257), would be more readable if organized in a table, perhaps commenting on the data.
In my opinion, the article is almost ready for publication in Materials, only minor revisions are needed.
best regards
Author Response
Dear Reviewer:
Thank you for your comments concerning our manuscript entitled “FeS2/C Nanowires as An Effective Catalyst for Oxygen Evolution Reaction by Electrolytic Water Splitting” (ID: materials-602695). Your comments are all valuable and very helpful for revising and improving our paper, as well as the important guiding significance to our researches. We have studied comment carefully and have made correction which we hope meet with approval. Revised portion are marked using the "Track Changes" function in Microsoft Word in the paper. The main corrections in the paper and the responds to the reviewer’s comments are as flowing:
1.Comments: Paragraph "3.2. Electrochemical oxygen evolution reaction" must begin with some text that introduces figure 6.
Response: Thanks for your comment. We have adjusted the position of figure 6 in the text, and in the minor revision, paragraph 3.2 begins with the text that introduces figure 6.
2.Comments: The comparison with other systems, whose specifications are reported in the text (lines 250-257), would be more readable if organized in a table, perhaps commenting on the data.
Response: Thanks for your suggestion. We have added a table entitled "Table S1 The comparison of catalytic performances for OER in 1 M KOH between the as-prepared FeS2/C nanowires and other materials reported in the literature." in the manuscript and supporting information.
We tried our best to improve the manuscript and made some changes in the manuscript. These changes will not influence the content and framework of the paper. And here we listed the changes. We appreciate for Editors/Reviewers’ warm work earnestly, and hope that the correction will meet with approval.
Once again, thank you very much for your comments and suggestions.
Best regards, Authors.
Reviewer 3 Report
The authors have addressed all the necessary comments as advised by the referees. Hence it can be accepted in its present form.
Author Response
Dear reviewer,
Thank you for your recognition of our work, which keeps us moving forward.
Once again, thank you very much for your comments and suggestions.
Best regards, Authors.